# The ORIGINS Project Biobank: A Collaborative Bio Resource for Investigating the Developmental Origins of Health and Disease

**DOI:** 10.3390/ijerph20136297

**Published:** 2023-07-04

**Authors:** Nina D’Vaz, Courtney Kidd, Sarah Miller, Minda Amin, Jacqueline A. Davis, Zenobia Talati, Desiree T. Silva, Susan L. Prescott

**Affiliations:** 1Telethon Kids Institute, North Entrance Perth Children’s Hospital, 15 Hospital Ave, Nedlands, WA 6009, Australia; courtney.kidd@telethonkids.org.au (C.K.); sarah.miller@telethonkids.org.au (S.M.); minda.amin@telethonkids.org.au (M.A.); jackie.davis@telethonkids.org.au (J.A.D.); zenobia.talati@telethonkids.org.au (Z.T.); desiree.silva@telethonkids.org.au (D.T.S.); susan.prescott@telethonkids.org.au (S.L.P.); 2School of Medical and Health Sciences, Edith Cowan University, Perth, WA 6027, Australia; 3School of Population Health, Faculty of Health Sciences, Curtin University, Perth, WA 6102, Australia; 4Department of Paediatrics and Neonatology, Joondalup Health Campus, Perth, WA 6027, Australia; 5Medical School, University of Western Australia, Nedlands, WA 6009, Australia; 6Scholars Program, Nova Institute for Health, Baltimore, MD 21231, USA; 7Department of Immunology, Perth Children’s Hospital, Nedlands, WA 6009, Australia

**Keywords:** ORIGINS, biobank, biorepository, birth cohort, non-communicable disease, microbiome, DOHaD

## Abstract

Early onset Non-Communicable Diseases (NCDs), including obesity, allergies, and mental ill-health in childhood, present a serious and increasing threat to lifelong health and longevity. The ORGINS Project (ORIGINS) addresses the urgent need for multidisciplinary efforts to understand the detrimental multisystem impacts of modern environments using well-curated large-scale longitudinal biological sample collections. ORGINS is a prospective community birth cohort aiming to enrol 10,000 pregnant people and follow each family until the children reach 5 years of age. A key objective is to generate a comprehensive biorepository on a sub-group of 4000 families invited to contribute blood, saliva, buccal cells, urine, stool, hair, house dust, cord blood, placenta, amniotic fluid, meconium, breastmilk, and colostrum over eight timepoints spanning the antenatal period and early childhood. Uniquely, ORIGINS includes a series of nested sub-projects, including interventions and clinical trials addressing different aspects of health. While this adds complexity as the project expands, it provides the opportunity for comparative studies. This research design promotes a multidisciplinary, multisystem approach to biological sample collection, analysis, and data sharing to ensure more integrated perspectives and solutions. This paper details the evolving protocol of our collaborative biobanking concept. Further, we outline our future visions for local, national, and ultimately international, comparative, and collaborative opportunities to advance our understanding of early onset NCDs and the opportunities to improve health outcomes for future generations.

## 1. Introduction

Non-Communicable Diseases (NCDs) pose a significant and escalating problem, with rates of obesity and associated cardiometabolic disorders, allergies, autoimmunity, cancers, and mental ill-health reaching epidemic levels [1]. According to The World Health Organization, this is driving the global burden of 41 million “premature” deaths each year, representing 74% of all deaths globally [2]. While attention has historically been placed on the treatment of established NCDs and/or on addressing risk factors in adulthood, there is increasing focus on the developmental origins of NCDs and the far greater benefits and higher impact of early life detection and prevention strategies [3]. Importantly, large scale biorepositories are critical to advancing the detailed understanding of biological mechanisms underlying disease development.

### 1.1. Development of the Origins of Health and Disease

The renowned microbiologist and environmentalist Rene Dubos pioneered the concepts of the “early origins” of health and disease in the 1960s [4,5,6]. He even predicted a pandemic of physical and mental disease as a delayed consequence of massive urbanisation, lifestyle, and environmental stress that would take generations to fully manifest—a prediction that is very much in evidence today [7]. However, while Dubos and his colleagues published many animal studies demonstrating the long-lasting influences of the early environment (maternal stress, nutritional deficiencies, and gut microbial alterations) on long term health [4,5,6,7], these concepts took many decades to gain traction. It was not until the 1990s that Barker proposed the Developmental Origins of Health and Disease (DOHaD) hypothesis after observing links between perinatal exposures, nutrition, birthweight and subsequent cardiovascular disease in humans [8]. DOHaD principles have since gained support through epidemiological studies and clinical interventions [9]. Additionally, utilising longitudinal biological sampling for mechanistic studies, it has become increasingly evident that early life exposures highly influence many aspects of metabolic, immune, and cognitive development during infancy and early childhood and thereby the susceptibility to a wide range of both early and later onset NCDs [10,11,12]. While genetic variation alone is not responsible for the rising NCD occurrence, gene-environment interactions are likely to affect both the occurrence and pathogenesis of NCDs [13] and should be considered in research designs.

### 1.2. Mechanistic Commonalities of NCDs

As discussed in our previous descriptions of this cohort [14], NCDs comprise a wide range of conditions and phenotypic variations but are all associated with chronic low-grade inflammation and immune dysfunction [7]. These common mechanistic threads have been more recently linked to underlying alterations of the gut microbiome [15,16,17]. This underscores the importance of shared risk factors and causal pathways in early life [18] and thereby also offers hope for multi-NCD intervention strategies implemented during prenatal and early life periods [19].

### 1.3. NCD Research and the Role of Biobanks

To unravel the origins of NCDs, there is a pressing need for well-designed, large-scale studies to investigate the biological complexities of the NCD epidemic and identify early life determinants. To achieve this, there is a requirement for substantial biorepositories with detailed complementary data collections, as well as collaborative multidisciplinary approaches between basic scientists, clinical carers, and policy makers.

Several large pre-conception, pregnancy, birth and early life biorepositories exist worldwide; these have provided valuable resources for robust epidemiological studies into health and disease [20,21,22,23,24]. As research platforms, these repositories contribute to the identification/classification of pregnancy and early life exposures and the development of strategies to optimise these for improved health outcomes.

Undoubtedly, advances in the areas of health and disease will be accelerated from more large-scale initiatives; this should be a priority either in the form of large individual studies or improved by the formation of consortiums with unified protocols to enable a comparative analysis. While large individual studies benefit from detailed data and sample collections, the increasing recognition of disease complexity has in more recent times demanded broader network-like study designs with increased collaborative and multidisciplinary components.

### 1.4. The ORIGINS Project and Biobank

The ORIGINS Project is an observational and interventional birthing cohort study aiming to recruit 10,000 birthing families from Perth, Western Australia (WA). As detailed elsewhere [14], ORIGINS is a research platform designed to enable investigations of early antecedent pathways to NCDs and how to curtail these. Through an integrated approach to the early environment (including parental physical health, biopsychosocial interactions of the ‘exposome’ with genetic disposition, and the developing microbiome as a major determinant of immune, metabolic, and emotional health) ORIGINS explores novel perspectives on promoting ‘a healthy start to life’. It provides an ideal opportunity to test new technologies and a rich evidence-base for policy makers and practitioners to enable healthier futures.

ORIGINS is a research platform that makes its Databank and Biobank resources available to researchers who undergo peer, scientific, and ethical review to become ‘nested’ sub-projects [25]. ORIGINS supports a core protocol for all participants and facilitates additional study-specific data and sample collections. All sample collections and subsequent experiments using the samples are performed in accordance with the National Statement on Ethical Conduct in Human Research [26]. Uniquely, the ORIGINS Biobank is able to adapt to the needs of sub-projects when required. This can include the real-time handling of specimens when sub-project protocol requirements differ from ORIGINS standard sample processing and cryopreservation.

The ORIGINS Biobank also embraces pre-emptive specimen handling to facilitate novel biotech applications and address cutting edge research areas as they arise. As a first example, the longitudinal maternal and infant stool from the antenatal period through to the child’s age of 5 years is intended to expedite rapidly developing research relating to the microbiome’s role in NCD development. As a second example, biological samples are being collected for future microplastic research with the methodology for quantification still in development.

The aims of the ORIGINS Biobank are to:Collect a comprehensive library of biological samples from ORIGINS participants to allow a large-scale cross sectional and longitudinal analysis.Process and store quality samples that facilitate sharing between the largest possible number of researchers and research fields.Facilitate cross-cohort centralised sample analysis (including genomic analysis) to optimise sample utilisation and achieve large data sets ideal for sharing.Engage and align with other large national and international birth cohorts and biobanks to facilitate comparative and collaborative initiatives on a local and global scale, ultimately advancing the understanding of NCD development and escalation.

## 2. Methods

### 2.1. Study Design

ORIGINS is both an observational and interventional population-based prospective cohort study, following participants from the antenatal period to early childhood. The eligible study population for ORIGINS is pregnant people who plan to deliver at Joondalup Health Campus (JHC) from both public and private wards; the majority of the study population resides in the Joondalup/Wanneroo area. Recruitment was initiated in January 2017 and, as of December 2022, over 8000 pregnant people have been enrolled. Ethics approval was granted by the Human Research Ethics Committee of JHC (ref 1440).

### 2.2. Study Area

ORIGINS is a multi-site study, with site locations spanning 46 km across Perth. Specimen collections take place in research clinics at JHC, on the wards of Joondalup Hospital, at the Joondalup community centre (implemented in 2022 as a response to COVID-19 restrictions), and in the homes of the participants in Joondalup, Wanneroo and surrounding areas. These samples are transported to ORIGINS laboratories located at the Telethon Kids Institute in Nedlands and to Western Diagnostics Pathology (WDP) facilities in Jandakot. The daily transportation of samples presents challenges and requires the involvement of reliable and dedicated medical and diagnostic couriers.

### 2.3. Recruitment

ORIGINS aims to recruit 10,000 pregnant people who plan to deliver at JHC, as well as their non-birthing partners and child (ren). Participants provide informed consent for the collection and long-term storage of data and/or biological samples, and for these to be released for use by ethically approved future studies. Participants are free to withdraw at any time and can opt to have their data and samples retained for future research or destroyed.

ORIGINS offers two levels of participation: ‘active’ and ‘non-active’ participation. This approach has allowed for the increased opportunity for data collection and opportunistic biological sample collection.

#### 2.3.1. Active Participants

Active participants provide informed consent to contribute biological samples as per Figure 1, complete online questionnaires regarding lifestyle and environment and allow access to their data from routine hospital data collections and data linkage from WA and the Australian Government. Active participants are also invited to join one or more sub-projects; specific consent is obtained for sub-project data and sample collections.

ORIGINS initially aimed to recruit 5000 active participants, but this target has been reduced to 4000 due to the impact of COVID-19. While the reduction in active participants has limited the numbers of biospecimens in the ORIGINS collection, the unique nature of samples and data collected during the pandemic period at least partially compensates for this reduction.

#### 2.3.2. Non-Active Participants

Non-active participants provide informed consent for ORIGINS to access their data from routine hospital data collections and data linkage from WA and the Australian Government. These participants also consent to the ORIGINS Biobank accessing and storing biological samples collected as part of routine care. Non-active participants may also agree to be contacted by sub-projects.

ORIGINS initially aimed to recruit 5000 non-active participants, but this target has been increased to 6000 to maintain a total of 10,000 participants across the two study arms.

### 2.4. Organisational Structure: Sub-Project Nesting

ORIGINS completes standardised data and biological sample collection activities at multiple defined timepoints, which are considered core collection activities. Uniquely, ORIGINS also facilitates additional sample and data collection/processing by approved nested sub-projects. These activities are subject to approval by governance and ethics committees and funding. Approved sub-projects are able to both leverage existing ORIGINS collections and create tailored protocols to facilitate novel research. They also add value to ORIGINS by returning their results back to the ORIGINS Databank and Biobanks to be used by future approved research.

Examples of special sub-project sample collection include the collection of infant blood and stool during the early post-natal period at 1 week, 3 months, and 6 months. There is also the special collection of amniotic cells harvested from amniotic membranes and the additional processing of infant blood samples prior to freezing to retrieve live mast cells. Finally, another sub-project is collecting nasal brushings from newborns and, using an integrative multi-omics approach, to examine how a vulnerable epithelium is related to respiratory health in infants. Protocols for these specialised sub-project collections are the intellectual property of the sub-projects and will not be discussed further here.

While these additional collections introduce complexity to the project, they add significant value to ORIGINS’ sample and data collections. Ultimately, this element of ORIGINS research design promotes the multidisciplinary and multisystem approach to samples collection and data sharing, which enables integrated perspectives on the DOHaD and pioneering solutions.

### 2.5. Biological Sample Collection

As shown in Figure 1, biospecimens are collected from the earliest recruitment timepoint at 20 weeks of gestation through to the child reaching 5 years of age. Biological samples are collected from pregnant people, non-birthing partners, and child participants. Table 1 outlines the sample types and volumes collected in the ORIGINS core protocol, the stored sample fractions, and the storage conditions.

### 2.6. Sample-Specific Procedures

ORIGINS employs specific procedures for the collection, processing, and storage of various biological samples. These procedures, including materials, transport, and storage, are detailed below.

#### 2.6.1. Bloods

Adult and child bloods are collected via venepuncture in research clinics in accordance with good clinical practice by trained phlebotomists and nurses. Bloods are kept at room temperature and delivered to the processing laboratory via diagnostic couriers within 4 h of collection. In 2022, a home-visiting phlebotomy service (Saturn Pathology) was engaged for blood home collection when requested by participants, resulting in an increased sample collection compliance. Saturn Pathology also assisted ORIGINS during COVID-19 restrictions and facilitated blood collections that would otherwise have been aborted. The details of the centrifuge programs used for blood processing are outlined in Table 2.

#### 2.6.2. Whole Blood

Whole blood is collected from 6 mL BD Vacutainer^®^ K2 Ethylene Diamine Tetra Acetic Acid (EDTA) tubes (Cat No. 367873, Becton Dickinson, East Rutherford, NJ, USA). A maximum of 2 × 1 mL aliquots are stored in FluidX tubes (Cat No. 68-1003-11, BioTools, Keperra, QLD, Australia). Aliquots are stored at −80 °C until requested for analysis by sub-projects.

#### 2.6.3. Plasma

EDTA plasma is collected from the 6 mL BD Vacutainer^®^ K2 EDTA tubes after whole blood has been collected. Lithium Heparin (LH) plasma is collected from 6 mL Becton Dickinson (BD) Vacutainer^®^ Heparin tubes (Cat No. 367885, Becton Dickinson). Tubes are spun once (EDTA = Program 5, LH = Program 4) and aliquoted into a maximum of 4 × 0.5 mL FluidX tubes (Cat No. 68-0703-11, BioTools). Aliquots are stored at −80 °C until requested for analysis by sub-projects.

#### 2.6.4. Serum

Serum is collected in 10 mL BD Vacutainer^®^ Serum Silica Clot Activator tubes (Cat No. 367895, Becton Dickinson, East Rutherford, NJ, USA). Tubes are spun once (Program 5) and aliquoted into a maximum of 6 × 0.5 mL FluidX tubes. Aliquots are stored at −80 °C until requested for analysis by sub-projects.

#### 2.6.5. PAXgene

Blood is collected into a 2.5 mL PAXgene^®^ tube (Cat No. 762165, PreAnalytiX, Hombrechtikon, Switzerland) containing a reagent that stabilises intracellular ribonucleic acid (RNA). The barcode on the label is scanned into a spreadsheet and the tube kept at room temperature until 2 h after collection. The original, labelled tubes are then stored at −80 °C until requested for analysis.

### 2.7. Peripheral Blood Mononuclear Cells (PBMC)

#### 2.7.1. Processing

Once plasma has been separated from the LH tubes as above, 20 mL of Gibco™ is mixed thoroughly and layered over 10 mL of room-temperature Lymphoprep™ (LP) density gradient (Cat No. 07851, STEMCELL Technologies™, Vancouver, BC, Canada). Tubes are spun (Program 1) and the resulting suspension of peripheral blood mononuclear cells (PBMC) is collected into a separate tube, topped up with Roswell Park Memorial Institute (RPMI), and spun again (Program 2). Supernatant is discarded and cell pellet resuspended in 10 mL of 2% solution of Heat Inactivated Fetal Calf (Bovine) Serum (HI-FCS) (Cat No. SFBS, Bovogen Biologicals Pty Ltd., Keilor East, VIC, Australia) and RPMI. Cells are spun (Program 3) and supernatant discarded. Depending on pellet size, it is resuspended in 4–10 mL of 2% HI-FCS/RPMI solution to create a minimum concentration of around 5 × 10^6^ cells/mL.

#### 2.7.2. Cell Counting

Solution is gently mixed and an aliquot is taken to dilute 1:1 with Gibco™ 0.4% Trypan Blue Solution (Cat No. 15250-061, Life Technologies, Carlsbad, CA, USA). A 10 uL aliquot is added to a slide specific for the TC20™ Automated Cell Counter (Cat No. 1450102, Bio-Rad, Hercules, CA, USA) to determine cell concentration and viability. The histogram lower gate is set to 6μm and the upper gate to 12μm. The total cell count is calculated as follows:Cell concentration (×10^6^) × resuspension media volume (mL) = Total PBMC × 10^6^

#### 2.7.3. Cell Storage

Cells are stored at a concentration of 10–20 × 106 cells/1 mL FluidX tube. Each tube contains 1 mL of solution composed of 50:50 cell suspension and freeze media (15% dimethyl sulfoxide (DMSO)/85% HI-FCS). FluidX tubes are pre-chilled in a 96 × 1 mL CoolRack (Cat No. BCS-149, BioTools).

#### 2.7.4. Cell Freezing

After counting, the tube is topped up with 2% HI-FCS/RPMI solution, mixed by inversion and spun on Program 3. Supernatant is discarded without disturbing the cell pellet. Cells are diluted in an appropriate volume of 2% HI-FCS/RPMI solution that is dependent on the number of cryovials being stored (see Table 3).

An equal volume of freeze solution is then added dropwise to the cell suspension over ice, or a 15 mL CoolRack (Cat No. BCS-153, BioTools). To reduce cell ‘shock’, the addition of the first 1 mL of freeze solution takes 1 min and can be quicker thereafter. The final solution is mixed well and 1 mL of suspension is added to each of the pre-chilled FluidX tubes. All tubes are moved to a CoolCell LX^®^ (Cat No. BCS-136, BioCision, LLC, Larkspur, CA, USA) and transferred to the −80 °C freezer immediately. After 24 h, the vials can be moved to liquid nitrogen for long term storage until required for analysis.

### 2.8. Adult and Child Buccal and Saliva Samples

Samples are collected via hypersalivation (drooling) and buccal swabs within the hospital research clinics by trained research assistants and nurses and kept cool until delivery to the processing laboratory within 4 h of collection.

#### 2.8.1. Saliva

Saliva samples are collected into sterile 5 mL tubes. Adult samples are aliquoted evenly using sterile transfer pipettes, between 1–3 × 0.5 mL FluidX tubes, with a minimum of 0.25 mL per tube. Child samples are also aliquoted evenly between 1–3 × 0.5 mL FluidX tubes with a minimum of 0.2 mL per tube. If only 0.1–0.2 mL is collected, the whole sample is collected into one tube. Samples are stored at −80 °C.

#### 2.8.2. Buccal Cells

Buccal cells are collected from both cheeks using a regular FLOQswab™ (Cat No. 502CS01, COPAN Diagnostics, Murrieta, CA, USA) and are immediately swirled in a sterile 5 mL tube containing 1 mL of RNAprotect^®^ Cell Reagent (Cat No. 76526, Qiagen, Hilden, Germany). The cell suspension is placed into a fridge for 24–72 h before using a sterile transfer pipette to aliquot evenly across 1–2 × 0.5 mL FluidX tubes, with a minimum of 0.4 mL per tube. Aliquots are stored at −80 °C.

### 2.9. Stool, Urine and Breastmilk

Stool, urine, and breastmilk samples are occasionally collected during clinic visits but are more commonly collected in the participants’ homes using provided instructions and collection kits with participant labels and barcodes. Stool samples are frozen and kept in the provided foam thermal container. Urine and breastmilk samples are kept cool by the participant by storing them in a refrigerator. Samples are brought to the hospital or a diagnostic branch by the participant or collected by a medical courier for transport to the laboratory. Urine and breastmilk samples must arrive to the lab on day of collection for processing but stool samples are accepted later than a day after collection, providing no freeze/thaw cycles have occurred. The implementation of medical couriers to assist with sample transportation from homes to the hospital has significantly improved compliance, especially in the early postnatal period when mothers are busy with infants and often find outings difficult. Meconium and colostrum are collected with help from nurses and midwives on the hospital wards and delivered via diagnostic couriers to the processing laboratories.

#### 2.9.1. Stool

In the early stages of the project, stool samples were collected by participants into 3 × 5 mL vials (Cat No. LBSSP2210X, ThermoFisher Scientific, Waltham, MA, USA) but participant consultation revealed significant logistical issues during collection, which affected compliance. Moreover, it was demonstrated that the microbiome composition of the 3 aliquots differed (1). After a change of protocol, stool samples began to be collected into a labelled 54 × 28 mm faeces tube containing a scoop (Cat No. 80.9924.014, Sarstedt, Nuembrecht, Germany). The sample is frozen by the participant and delivered frozen to the lab in a provided thermal container (Cat No. 95.1011, Sarstedt), then weighed and stored in the original tube at −80 °C until requested for analysis. At the first request, stool samples are homogenised and split into barcoded tubes, and an aliquot of homogenised sample is sent to requesting researchers.

#### 2.9.2. Meconium

Meconium is collected into the same containers as stool samples (Cat No. 95.1011, Sarstedt) and sent to the laboratory, where it is frozen immediately at −80 °C. It is noted whether the meconium is believed to be the first meconium passed by the newborn.

#### 2.9.3. Urine (Adult)

Samples are identified as being a first void or random sample and stored within 36 h of collection. The colour and turbidity of the sample is noted. The collection pot is wiped with 70% ethanol before opening the lid. Using a sterile transfer pipette, 1.8 mL of urine is firstly aliquoted into a 1 × 2 mL FluidX glass tube (Cat No. 65-9001, BioTools) and 4 × 2 mL plastic FluidX tubes thereafter. Tubes must have a minimum of 0.5 mL each. Aliquots are stored at −80 °C.

#### 2.9.4. Urine (Child)

Child urine samples must be stored within 36 h and are not processed if faecal contamination is present. Child samples may contain urine or cotton wool balls soaked with urine. Both sample collection containers are wiped with 70% ethanol before opening. If containing cotton wool, the balls are removed with tweezers cleaned with 70% ethanol and placed into a 10 mL or 20 mL sterile syringe with the plunger removed. The plunger is replaced and depressed so that the urine is collected back into the original pot. The colour and turbidity of the sample is noted before aliquoting and storing as per adult samples, above. If there is insufficient volume to fill all 5 FluidX tubes, a minimum of 0.5 mL is aliquoted into as many tubes as possible.

#### 2.9.5. Colostrum

Samples arrive in 1–2 × 1 mL labelled syringes. Syringes are wiped with 70% ethanol before expelling the contents into a separate, non-sterile container and mixing thoroughly. Any unusual colouring is noted. 500 µL is aliquoted into as many 0.5 mL FluidX tubes as possible. Aliquots are stored at −80 °C until required for analysis.

#### 2.9.6. Breastmilk

Samples are stored within 36 h and any unusual colouring noted. The collection pot is wiped with 70% ethanol before opening. Breastmilk is thoroughly mixed with a sterile transfer pipette before firstly aliquoting 1.8 mL into a 1 × 2 mL FluidX glass tube and 8 × 2 mL plastic FluidX tubes thereafter. If there is minimal volume, 1 mL of sample is aliquoted into as many tubes as possible. Tubes are stored at −80 °C until requested.

### 2.10. Hair Samples

Hair samples are collected in research clinics by trained research assistants and nurses and stored at room temperature. Approximately 5 cm of 100 hair strands is collected close to the scalp into labelled borosilicate glass culture tubes (Cat No. LBSDCT1275, ThermoFisher Scientific), catalogued and stored at room temperature until requested for analysis by sub-projects. It is important the hair is stored in a consistent direction in the culture tubes, as the hair strand is essentially a timeline with most recent exposures/biological deposits reflected closest to the scalp.

### 2.11. Teeth

As child participants start to reach the age where they will lose their baby teeth (around age 6), a protocol will be established for the collection and storage of these samples. The planning for this protocol is currently underway.

### 2.12. Birth Samples

Cord bloods and placentas are collected in the birthing wards by midwives and obstetricians and couriered to the processing laboratory at the first opportunity. Samples delivered in the late afternoon and evening are delivered to the laboratories the following morning. Medical Couriers have been imperative to this process.

#### 2.12.1. Amniotic Fluid

Amniotic fluid has been collected from a small number of Caesarean sections via a clean catch method into a sterile kidney dish, from where the fluid is transferred to sterile heat-treated glass jars. Amniotic fluid collection is only performed by a small number of private obstetricians and these collections are under strict ethical control to ensure there is no compromise to the health and wellbeing of pregnant people and infants. Amniotic fluid samples travel on ice with other birth samples to the processing laboratories in the shortest timeframe possible. Samples are transferred from glass collection jars using glass pipettes to 10 × 2 mL glass FluidX storage vials. Glass labware is used for these samples, as it is the intention to measure plastic chemicals in the amniotic fluid; the potential for plastic contamination should be minimised. Samples are stored at −80 °C.

#### 2.12.2. Cord Blood Mononuclear Cells (CBMCs)

Cord blood is collected from the umbilical cord vein after the baby has been born and the cord has been clamped and cut. The vein must be swabbed with alcohol and maternal blood gently removed before the collection of the sample using an 18 Gauge SafetyGlide™ needle (Cat No. 305918, Becton Dickinson) and a 30 mL syringe. The gauge of the needle limits cell damage due to shredding. Blood is transferred into all necessary tubes through the syringe or other transfer device; pouring between tubes must not occur in order to prevent cross-contamination of tube preservatives. The majority of the sample is transferred into a room-temperature 50 mL conical tube containing 20 mL of Gibco™ RPMI 1640 Medium and 500 µL of Heparin Sodium (Cat No. AUST R 49232, Pfizer, New York, NY, USA) to support cell survival. As cord blood tends to have more red blood cell contamination than peripheral blood collections, two Lymphoprep™ density gradient steps are required.

#### 2.12.3. Cord Blood Mononuclear Cells (CBMC) Processing

The RPMI/blood mixture volume is divided equally and gently layered over 2 × 10 mL Lymphoprep™ density gradients at room temperature in 50 mL centrifuge tubes. Any large blood clots are left in the original tube. Both LP tubes are spun on Program 1 and the resulting cloudy layer of cord blood mononuclear cells (CBMCs) is collected into a separate 50 mL tube and topped up with RPMI. The tube is spun in a cell wash step on Program 2 and the supernatant is discarded without disturbing the cell pellet. Small pellets are resuspended in 5 mL of RPMI and gently layered over 2.5 mL of Lymphoprep™. Alternatively, large pellets are resuspended in 12.5 mL of RPMI and layered over 15 mL of Lymphoprep™. The LP tube is spun on Program 1 for only 20 min before again collecting the layer of CBMCs into a separate tube, topping up with RPMI, and centrifuging on Program 2. Supernatant is discarded and cell pellet resuspended in 1–2 mL of 2% HI-FCS/RPMI solution before transferring to a 15 mL conical tube and topping up to 10 mL. The tube is spun (Program 3) and supernatant is discarded. Depending on pellet size, it is resuspended in 4–10 mL of 2% HI-FCS/RPMI solution to create a minimum concentration of around 5 × 105 cells/mL.

#### 2.12.4. Cell Counting and Freezing

As above for PBMCs.

#### 2.12.5. Granulocytes

Cells are collected from the top of the red blood cell layer (velvety in appearance) after the first LP spin. They appear as small, white specks on top of the layer and are collected by a ‘dabbing’ approach using a sterile, transfer pipette. A total of 1 mL of sample is collected into a 1 × 1 mL FluidX tube and stored at −80 °C. However, if the EDTA whole blood was unable to be collected, then a maximum of 2 × 1 mL FluidX tubes are stored.

#### 2.12.6. Placenta

Complete placentas are cleaned to remove excess blood, clots, and meconium deposits, then weighed and photographed from the maternal and foetal side, along with a side profile to show tissue thickness. A metal ruler is used in the picture for scale. The following observational data are recorded for all placentas: cord insertion type, presence of calcifications, meconium staining/deposits, and whether the tissue is bilobed. For placentas <48 h old, membrane and tissue samples are collected. Using a sterile scalpel blade, a 5 × 5 section of full thickness membrane (intact amnion and chorion) is cut with the rupture site at the top of the section and then equally divided into 3 horizontal layers to capture all membrane areas: the rupture site (M-R), middle section (M-M), and basal plate/placental margin (M-B). Each layer is further sub-divided into 3 squares before adding all pieces into their respective 2 mL plastic FluidX tubes, labelled as M-R, M-M, or M-B.

Tissue sections must capture the full thickness of the placenta and are collected using an 8 mm disposable, dermal biopsy punch (Cat No. KAI-706, Kai Medical, Honolulu, HI, USA). The placenta is placed cord side up before taking 6 samples from the full width of the placental disc. Samples are representative of the whole placenta—tissue abnormalities are avoided if possible. The top membrane layer is discarded, then each segment equally divided into 3 horizontal sections: the villus-core (P-VC), villus (P-V), and decidua (P-D). Samples are placed into the labelled 2 mL plastic FluidX tubes relevant to that section before all membrane and tissue aliquots are snap frozen in liquid nitrogen. Samples are then transferred to −80 °C storage until required for analysis.

### 2.13. Dust

Dust is collected straight from participants’ vacuum cleaners in zip lock bags, delivered to the study staff during visits and stored at −20 °C until requested for analysis by sub-projects. The processing/sifting of dust samples is currently in progress.

## 3. Results

### 3.1. Biospecimen Compliance

#### 3.1.1. Antenatal Clinic Sample Collections

Up to a 95% compliance rate has been observed with the antenatal sample collections. Some limitations occurred during COVID-19 restrictions and occasionally sample refusal or collection difficulties prohibit sampling.

#### 3.1.2. Birth Ward Collections

Approximately 75% of cord blood and placenta samples have been collected. Limitations relate to private stem cell collections, clinical priorities at the birth, and diagnostic requirements for the placental tissues, all of which take priority over the use of these samples and tissues for research.

#### 3.1.3. Child Clinic Sample Collections

Up to a 60% compliance rate has been observed with child sample collections in clinic, but compliance varies. Limitations are significant and include parent refusal and difficult sample collections.

#### 3.1.4. Home Sample Collections

Compliance rates of around 60–80% are observed for these samples when aligned with a clinic appointment and around 30–40% in the postnatal period when there is no alignment with clinic attendance. The main limiting factor relates to the early postnatal period when parental occupation with the infant is significant. Home couriers have improved compliance with samples collected in the home, which partly mitigates limitations (detailed below).

### 3.2. Post-Collection Procedures

#### 3.2.1. Real Time Diagnostic Results

Participants consenting to paediatric bleeds at the ages of 1, 3, and 5 years are offered a diagnostic full blood count and ferritin analysis. These diagnostic results are communicated to participants by a paediatrician, along with any warranted medical advice. The feedback of results has proven to be a strong incentive for participants consenting to child bleeds and increasing compliance. This diagnostic data has also been useful for several sub-projects and has thus been extremely valuable to the project overall.

#### 3.2.2. Request and Release of Biospecimens

Requests for ORIGINS biological samples are proposed by sub-projects, which are initially assessed by the Biobank Manager. Sub-projects then progress to the review and approval by a relevant Research Interest Group, ORIGINS Scientific Committee and the Ramsay Health Care Human Research Ethics Committee, before proceeding to implementation. At the point of sample analysis, a detailed request for the release of specific biological samples is completed. This request confirms the analysis being undertaken, the analysis timeline, the facility that will undertake the analysis and research, the expertise level of the team and analysis facility and, importantly, the commitment to provide any derived data back to the ORIGINS Databank for future use. When possible, any excess or derived biological samples are returned to the ORIGINS Biobank.

#### 3.2.3. Request and Release of Biospecimen Metadata

The ORIGINS Biobank holds metadata related to all biospecimens, including the time of collection, processing and freezing, colour and clarity of fluids, degree of blood haemolysis, sample observations, and any cautions/considerations for use, such as known illnesses at the time of collection, including COVID-19.

#### 3.2.4. Request and Release of Participant Data

The ORIGINS Databank holds participant data, which is requested via the same approval procedures as biospecimens and overseen by the Databank Manager. Participant data such as health information is collected from multiple sources including hospital records and questionnaires, as discussed elsewhere [14].

### 3.3. Validation of Sample Specific Procedures and Current Use of Samples

ORIGINS blood collection protocols have been utilised in our laboratories for many years and yield viable cryopreserved PBMC and CBMC populations, which have so far been successfully utilised for live cultures and the immune characterisation of adults and children. Plasma and serum samples have been utilised for metabolomic, inflammation, and antibody assays. Stool samples have mainly been utilised for microbiome analysis, and urine has proven useful for metabolomic analysis. Whole blood samples have been utilised for DNA extraction, with buccal samples being valuable backups when blood is not available. Placenta samples have so far been used for the analysis of amnion cells. Hair is being used to analyse cortisol levels, and breastmilk and dust are currently being analysed for the allergen content in relation to allergic disease. ORIGINS saliva samples have not yet been requested by approved sub-projects; we believe these would be of a higher value if collected from fasting individuals, which was not feasible in this study due to clinical visits occurring throughout the day.

### 3.4. Addressing and Overcoming Challenges

#### 3.4.1. COVID-19 Impact on Biobanking

While WA has, to date, avoided lengthy COVID-19 lockdowns, several brief lockdowns and periods of social restrictions brought some interruptions to biobanking and compliance rates. During lockdowns and times of increased social distancing, virtual appointments replaced on-site clinic visits, which prevented blood, saliva, and buccal collection. ORIGINS aimed to increase compliance through home sample collection by a mobile blood service, and by increasing on-site sample collection after the cessation of restrictions; however, compliance remained lower than during the pre-COVID-19 period. Birth collections of cord bloods and placentas continued largely uninterrupted during lockdown and restrictions.

While clinic-collected samples were reduced, there was actually an increase in the engagement and compliance for home-collected samples (e.g., urine, stool) during COVID-19 restrictions. Participants reported increased availability of time for sample collections and an increased commitment to research during a time of general health-related uncertainty.

The availability of two complementary processing sites has proven advantageous during COVID-19 lockdowns, allowing staff to socially distance and operate in isolation if either site was affected by a virus exposure. While the location of the ORIGINS clinical facilities in the Joondalup Hospital have generally been advantageous, this was not the case during COVID-19 outbreaks as restrictions were established in clinical environments. These restrictions in clinical environments were mitigated successfully with the implementation of offsite clinic locations, but the project experienced unavoidable delays while establishing new facilities.

#### 3.4.2. Biobank Data Management

The handling of sizeable biobank data requires a comprehensive Laboratory Information Management System (LIMS). While ORIGINS was initiated with limited data management resources, OpenSpecimen (OS) software (version 7.0, Krishagni Solutions Pvt Ltd, Pune, Maharashtra, India) was purchased and implemented in 2018. Legacy data was imported into OS and a significant customisation process, including integration with REDCap. This integration enabled the software to manage the complex longitudinal design of ORIGINS that includes several levels of conditional logic to accommodate multiple consent levels and customised sub-project sample collections. Where possible, it is highly recommended that a comprehensive data strategy and software optimisation is developed prior to the undertaking of biobanking research.

#### 3.4.3. Biospecimen Cost Calculations

Biobanks commonly experience difficulty achieving a viable cost recovery system while retaining competitive pricing for researchers who often have limited budgets [27,28]. Due to core funding, the ORIGINS Biobank does not rely on a full cost recovery model; however, partial cost recovery is essential to assist with the continuation of the Biobank and the wider project. With this in mind, ORIGINS and Biobank management staff apply for infrastructure grants at every opportunity, and recover funds through the per-sample costing for approved sub-projects. The cost of biological samples to sub-projects considers:(1)Scale of the proposed project: larger projects are encouraged by an inverse cost/scale calculation; it is the intention to prioritise very large-scale studies where possible to achieve a completeness of data.(2)Abundance of sample type: cost increases with rarity, which particularly affects infant and child samples that are difficult to obtain and collected in small volumes. Very rare samples, and especially those of which there is only one, are not currently being released without very careful consideration by ORIGINS Management and the Executive Committee.(3)Study designs: longitudinal designs incur a cost increase per additional timepoint, as these sample sets are relatively rare and expensive to collect due to the continued engagement and retention effort required.(4)Investigator affiliation: academic investigators, particularly those from Telethon Kids Institute and JHC, pay reduced fees for samples. In comparison, industry investigators incur higher fees on the assumption that industry projects are intended for commercial purposes.(5)Funding limitations are assessed and accommodated on a project-to-project basis, where possible.

#### 3.4.4. Outreach Actions of the ORIGINS Biobank

To raise awareness of the ORIGINS Biobank and biospecimen collections, all opportunities are taken to attend and present at meetings and conferences locally, nationally, and internationally. There is also ongoing engagement with local and national media to present research findings. As the cohort matures and sample collections reach completion, there are plans to increase advertisement activities.

## 4. Conclusions

The ORIGINS Biobank is a continually evolving biorepository designed with some commonalities to other biobanks including the Raine study [29], the Gen R study [24], and the ALSPAC study [23]. The ORIGINS Biobank offers specialised collections for the sub-project, similar to Environmental influences on Child Health Outcomes (ECHO) Program [22].

A central objective of the ORIGINS Biobank is, in tandem with the ORIGINS Databank, to provide a platform to engage local, national, and international clinicians; researchers; the government; and the wider community to facilitate advancement and translational outcomes in the context of NCDs.

The ORIGINS research platform currently (as of early 2023) supports 42 sub-projects, of which 25 take advantage of the ORIGINS Biobank in both real-time and retrospective designs. With several real-time sub-projects involving sample collections, the maintenance of concurrent core and sub-project sample collections present significant challenges for the Biobank. This complex structure has been made possible by establishing a research laboratory with specialised staff that undertake specialised and time-consuming processes, as well as by the close collaboration with WPD that has the capacity for challenging, high throughput workflows. This close collaboration has also facilitated real-time diagnostic results, which has proven to be an effective incentive for families to complete the biological sampling, delivering valuable findings for both the research and clinical follow up. Overall, the partnership between clinical research and diagnostic providers has proven mutually beneficial/valuable and a model worth considering for other large-scale biobanks and diagnostic providers wishing to develop a research profile.

ORIGINS’ multi-site design has proved challenging. Samples collected at multiple locations (JHC, Joondalup Hospital, the community centre, and at participant’s homes) need to be regularly and quickly transported to laboratories located more than 45 km away. The daily transport of samples requires the involvement of reliable and dedicated medical and diagnostic couriers that incur significant costs.

While ORIGINS is designed to enable a strategic long-term research capacity, it has also been a goal to create a ‘responsive’ system with ‘real-time’ feedback to parents and their children, and translation to clinical and diagnostic services. Essentially, ORIGINS challenges the classic research ‘watch and wait’ cohort design, where the documentation of outcomes without interference is the end goal. While ORIGINS is not designed to watch ill-health trajectories run their course to objective endpoints, the ability to make a difference for young families and contribute to a compassionate community approach fits the project objective of supporting healthy communities.

Longer term, ORIGINS ultimately aims to facilitate a large cross-cohort longitudinal multiomic analysis, which, in combination with exposome measures, will assist the understanding of the complex interactions that form the foundations of health and disease in the early years of life. As an example, the cross-cohort metabolomic analysis is in planning stages to investigate the maternal, paternal, and child metabolome in the context of health and disease in both parents and offspring. Large scale studies of the genome, epigenome, transcriptome, and virome are examples of other future initiatives, which will be formalised as the cohort matures. To optimise a transgenerational perspective, The ORIGINS Project is planning the recruitment of grandparents and the analysis of biological samples from three generations, where logistically possible.

The core processes (cohort maintenance, data collection, and sample collection) of The ORIGINS Project required large scale funding, which was secured in 2016 for a 10-year duration. This core funding, however, was intended for data collection only and all analysis relies on additional funding applications through approved sub-projects. Additionally, a Biobank cost recovery plan to ensure the longevity of the sample and data repositories beyond the initial 10-year funding period has been developed but, like for most biobanks, the cost recovery is only partial. Factors affecting cost recovery include suboptimal research funding and finding an often-difficult balance between facilitating underfunded research projects and protecting financial interests of the Biobank. To offset research ‘losses’, The ORIGINS Project is contemplating collaborations with industry, but a framework for commercial interactions that is acceptable to all parties, including participants and consumers, is in early stages and has not yet been finalised.

Collaborations locally, nationally, and internationally are a high priority for ORIGINS, which requires a complex Biobank in order to facilitate large scale and regional comparative studies. There are currently several initiatives developing that will compare ORIGINS samples and data to other birth cohorts, including the GenV study in Victoria, Australia and The Born in Bradford study in the United Kingdom. A very exciting regional opportunity is developing between The ORIGINS Project and two large, internationally renowned West Australian longitudinal cohort studies, the Busselton Health Study [30] and the Raine Study [29]. This collaboration will allow regional research spanning almost 60 years of biological history in WA.

The ORIGINS Biobank encompasses comprehensive longitudinal collections of biological samples that are available to researchers as part of real-time or retrospective project designs. The large cohort size and inclusion of specialised sub-projects has generated complex high throughput biobanking workflows, which have significantly benefitted from the collaboration with diagnostic service providers and the incorporation of real-time diagnostic testing and feedback to participants. Ultimately, it is the goal of The ORIGINS Project to facilitate multidisciplinary research to help elucidate the early origins of NCDs and support the maintenance of healthy communities and healthier environments across local, national, and global scales.

## Figures and Tables

**Figure 1 ijerph-20-06297-f001:**
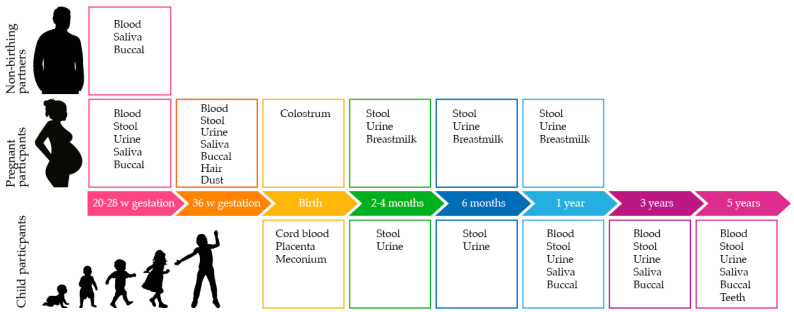
Biological samples provided by non-birthing partners, pregnant participants, and child participants at each timepoint.

**Table 1 ijerph-20-06297-t001:** Sample types, volumes, and storage conditions of ORIGINS core biological samples.

Sample	Collection Volume	Stored Material	Storage Vessel	Storage Temp
Adult Blood	30 mL	Plasma, serum, whole blood, granulocytes, RNA, PBMC	Plastic	−80 °C, PBMC −197 °C
Cord Blood	60 mL	Plasma, serum, whole blood, granulocytes, RNA, CBMC	Plastic	−80 °C, CBMC −197 °C
Child Blood	10 mL	Plasma, whole blood, Granulocytes, RNA, PBMC	Plastic	−80 °C, PBMC −197 °C
Placenta	Whole	Snap frozen membrane biopsies, tissue biopsies, cotyledons	Plastic and glass	−80 °C
Saliva	1 mL	Saliva	Plastic	−80 °C
Buccal Swab	1 swab	Buccal	Plastic	−80 °C
Urine	60 mL	Urine	Plastic and glass	−80 °C
Stool	60 mL	Stool	Plastic	−80 °C
Meconium	60 mL	Meconium	Plastic	−80 °C
Colostrum	2 mL	Colostrum	Plastic	−80 °C
Breast milk	60 mL	Breastmilk	Plastic and glass	−80 °C
Hair		Hair	Plastic	Room temp
Dust	1 bag	Dust	Plastic	−20 °C
Teeth	1	Teeth	Plastic	−20 °C

RNA = ribonucleic acid, PBMC = peripheral blood mononuclear cells, CBMC = cord blood mononuclear cells.

**Table 2 ijerph-20-06297-t002:** Centrifuge programs for blood processing.

	Program 1	Program 2	Program 3	Program 4	Program 5
Temperature (°C)	18 °C	18 °C	18 °C	18 °C	18 °C
Speed (rpm)	1580	1580	1580	2000	4000
Time (min)	30	10	7	10	10
Acceleration	5	9	9	9	9
Deceleration	0	9	9	9	9

**Table 3 ijerph-20-06297-t003:** Peripheral blood mononuclear cell (PBMC) counts, aliquot number, cell pellet suspension, and volume (Vol). Heat inactivated Fetal Calf Serum (HI-FCS). Roswell Park Memorial Institute (RPMI) Medium. Dimethyl sulfoxide (DMSO).

PBMC Count (×10^6^)	Number of Cryovials	Vol (mL) 2% HI-FCS/RPMI	Vol (mL) 15% DMSO Freeze Solution
10–20	1	0.5	0.5
21–40	2	1.0	1.0
41–60	3	1.5	1.5
61–80	4	2.0	2.0

## Data Availability

No new data were created or analyzed in this study. Data sharing is not applicable to this article.

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
