# Peer review of "The ORIGINS Project Biobank: A Collaborative Bio Resource for Investigating the Developmental Origins of Health and Disease"

_ijerph, 2023, doi:10.3390/ijerph20136297_

Round 1
Reviewer 1 Report
D'Vaz et al. present a detailed description of a biobank project. Particularly, I found the manuscript relevant and easily readable. I have some suggestions for the authors:
1. It would be interesting if the authors could comment more on NCD being multifactorial diseases, explaining the genetics and environment interaction, especially because the biobank samples could be used to access both these features.
2. I suggest the authors add a brief explanation regarding the uses and applications of the samples collected.
3. Are there any studies already being applied with ORIGIN biobank samples?
4. Table 1 is not well configured. I believe it would benefit the manuscript if the authors transform this table into a figure, as a timeline, with the individuals and their samples.
5. Small suggestion: cord blood samples could be placed next to peripheral blood samples.
Reviewer 2 Report
The manuscript describes a very comprehensive collection of samples, being an example of alliance between clinicians, researchers and participants collaborating with the Biobank. Although the aims of the Biobank are listed and protocols for collection of samples are detailed in depth, other critical information is missing:
· Associated data included in the databank.
· Limitations of the recruitment for the usefulness of the collection.
· Justification of the study design for future applications.
· Validation of sample-specific procedures and expected usefulness of the obtained derivatives.
· Considerations about informed consents and the potential broad use of samples and data.
· Outreach actions for the collection.
· Cost recovery model.
· Commonalities to other biobanks.
Other minor issues are:
- The date of beginning of the enrolment is not indicated in the manuscript.
- Sample-specific procedures don’t include number of blood tubes collected, number of aliquots of derivatives stored, or centrifugation settings.
- Additional references should be included for the whole manuscript and specifically, in relation with 3.3. section.
- The organization of the manuscript should be revised: results are presented as methods (2.14 and 2.15) or conclusions (supported projects), whereas Results section is missing.
- Abbreviations should be collected in a specific section.
- Bullet points should be used on lines 117-125.
- Tables format could be improved, and the key locations marked in Figure 1.
